

# New free-living nematode species and records (Chromadorea: Plectida and Desmodorida) from the edge and axis of Kermadec Trench, Southwest Pacific Ocean

Daniel Leduc

National Institute of Water and Atmospheric Research, Wellington, New Zealand

## ABSTRACT

One new nematode species is described and two new species records are provided from the edge (6,080 m depth) and axis (7,132 m) of Kermadec Trench, Southwest Pacific. *Leptolaimus hadalis* sp. nov. is characterised by medium body 587–741 μm long, labial region not offset from body contour, inconspicuous labial sensilla, amphid located 12–19 μm from anterior end, female without supplements, male with four tubular precloacal supplements (alveolar supplements absent), tubular supplements almost straight with dentate tip, arcuate spicules and weakly cuticularized dorsal gubernacular apophyses strongly bent distally. In a previously published ecological survey of Kermadec Trench, *L. hadalis* sp. nov. was the most abundant species in a core obtained at 8,079 m water depth and third most abundant species in a core obtained at 7,132 m, while only one individual was found at 6,096 m depth, and none at 9,175 m depth (*Leduc & Rowden, 2018*). *Alaimella aff. cincta* and *Desmodora aff. pilosa* are recorded for the first time from the Southwest Pacific region. Prior to the present study, *Alaimella* had only been recorded from coastal locations and from the Weddell sea to a depth of 2,000 m. The record of *Desmodora aff. pilosa* at 6,080 m depth is the deepest record of a *Desmodora* species to date, although unidentified *Desmodora* specimens have been found as deep as 6,300 m in the South Sandwich Trench. The morphology of the Kermadec Trench *Alaimella aff. cincta* and *Desmodora aff. pilosa* specimens bear a strong resemblance to their respective type populations from the Northern Hemisphere, but further morphological and molecular data are required to ascertain whether they in fact represent distinct species.

# INTRODUCTION

Unlike most other metazoans, nematodes communities in hadal trenches (>6,000 m depth) are characterized by relatively high abundances and species richness despite extreme conditions (*Zeppilli et al., 2018*). A study of three hadal sites in Puerto Rico Trench (Northeast Atlantic) revealed the presence of almost 150 morphospecies (*Tietjen, 1989*),

Corresponding author
Daniel Leduc, daniel.leduc@gmail.com

whilst 109 morphospecies were identified from four sites in Kermadec Trench in the Southwest Pacific (*Leduc & Rowden, 2018*), and 36 morphospecies were identified in cores from Tonga Trench's deepest point (Southwest Pacific; *Leduc et al., 2016*).

In order to understand patterns and drivers of benthic diversity within and across trench habitats, it is necessary to better describe the hadal nematode fauna. However, most of these hadal species are yet to be described and named. In their review of deep-sea nematode taxonomy, *Miljutin et al. (2010)* showed that only 10 named species had so far been recorded from >6,000 m depth, compared to 210 named species from abyssal plains (4,000–6,000 m depth). The same authors listed 46 bathyal and/or abyssal nematode species which are found in more than one ocean, suggesting a cosmopolitan distribution. *Leduc & Rowden (2018)* have shown that the adjacent Kermadec and Tonga trenches share 33 morphospecies, however no other studies have yet compared the nematode species composition between trenches, and it is not clear whether any hadal nematode species has a wide geographical distribution within or across ocean basins. In order to facilitate comparison of hadal nematode fauna across trenches, detailed morphological descriptions of one new species and two new species records are provided from the edge and axis of Kermadec Trench.

## MATERIALS & METHODS

The Kermadec Trench extends from approximately 26 to 36°S near the northeastern tip of New Zealand, Southwest Pacific. Sediment samples were obtained from the edge of Kermadec Trench at 6,080 m depth during *RV Tangaroa* cruise TAN1711 in December 2017 using a USNEL-type box corer (dimensions: $0.5 \times 0.5 \times 0.5$ m, 0.125 m$^3$ capacity). Subsamples were obtained using a cut-off syringe with 29 mm internal diameter to a depth of 10 cm, sliced into 1 cm layers and fixed in 10% buffered formalin. Additional samples were collected from the axis of Kermadec Trench at 7,132 and 8,079 m depth during Woods Hole Oceanographic Institute (WHOI) cruise TN309 (*RV* Thomas G Thompson) in May 2014 (*Leduc & Rowden, 2018*). The sediment cores were obtained using the submersible Nereus (core internal diameter = 6.35 cm). The cores were sliced into 0–1, 1–2, 2–3, 3–4, 4–5 and 5–10 cm layers and fixed in 10% buffered formalin. Samples were rinsed on a 20 μm sieve using freshwater, nematodes extracted from the remaining sediments by Ludox flotation, stained with Rose Bengal, and transferred to pure glycerol (*Somerfield & Warwick, 1996*).

Species descriptions were made from glycerol mounts using differential interference contrast microscopy and drawings were made with the aid of a camera lucida. Measurements were obtained using an Olympus BX53 compound microscope with cellSens Standard software. All measurements are in μm, and all curved structures are measured along the arc. The terminology used for describing the arrangement of morphological features such as setae follows *Coomans (1979)*. Type specimens are held in the NIWA Invertebrate Collection (Wellington). The collection of sediment samples was conducted under Special permit 666 to NIWA granted by New Zealand's Ministry for Primary Industries.

The electronic version of this article in Portable Document Format (PDF) will represent a published work according to the International Commission on Zoological Nomenclature (ICZN), and hence the new names contained in the electronic version are effectively published under that Code from the electronic edition alone. This published work and the nomenclatural acts it contains have been registered in ZooBank, the online registration system for the ICZN. The ZooBank LSIDs (Life Science Identifiers) can be resolved and the associated information viewed through any standard web browser by appending the LSID to the prefix http://zoobank.org/. The LSID for this publication is: urn:lsid:zoobank.org:pub:47EF84F2-7B80-460E-BAF0-747FA9DDD447. The online version of this work is archived and available from the following digital repositories: PeerJ, PubMed Central and CLOCKSS.

## Systematics

Order Plectida *Gadea, 1973*
Family Leptolaimidae *Örley, 1880*
Genus *Leptolaimus de Man, 1876*

**Generic diagnosis: (from *Holovachov (2014)*)** Lateral alae present. First annule anterior to cephalic setae bases and amphids, cephalic capsule absent. Cephalic sensilla papilliform or setiform. Amphideal aperture ventrally unispiral, without central elevation. Secretory-excretory system present; excretory canal short, excretory ampulla present. Ovary branches reflexed antidromously. Alveolar or tubular supplements present in females of some species, either in pharyngeal or pre-anal regions, or in both positions. Male reproductive system diorchic. Number of supplements varies from zero to 40 for alveolar and zero to 11 for tubular; males may have both types of supplements, one or no supplements at all. Caudal glands and spinneret present or absent.

Type species: *L. papilliger de Man, 1876*

**Remarks.** The genus was revised by *Holovachov & Boström (2013)* who provided a key to the identification of all 58 valid species based on features of both males and females. Three species were subsequently described by *Tchesunov (2015)*, *Qiao, Jia & Huang, 2020* and *Leduc (2020)*.

### *Leptolaimus hadalis* sp. nov.

Figures 1–3, Table 1
urn:lsid:zoobank.org:act:892D1646-DD68-4601-A12C-BA26A88E1B1A

**Type locality:** Kermadec Trench, 7,132 m water depth, sediment depth 0–2 cm, *RV Thompson* voyage TN309, stations N073, 35.8396°S, 178.879°W.

**Type material**: Holotype male (NIWA 154899), and three paratype males and three paratype females (NIWA 154900), collected in May 2014.

**Measurements:** See Table 1 for detailed measurements.
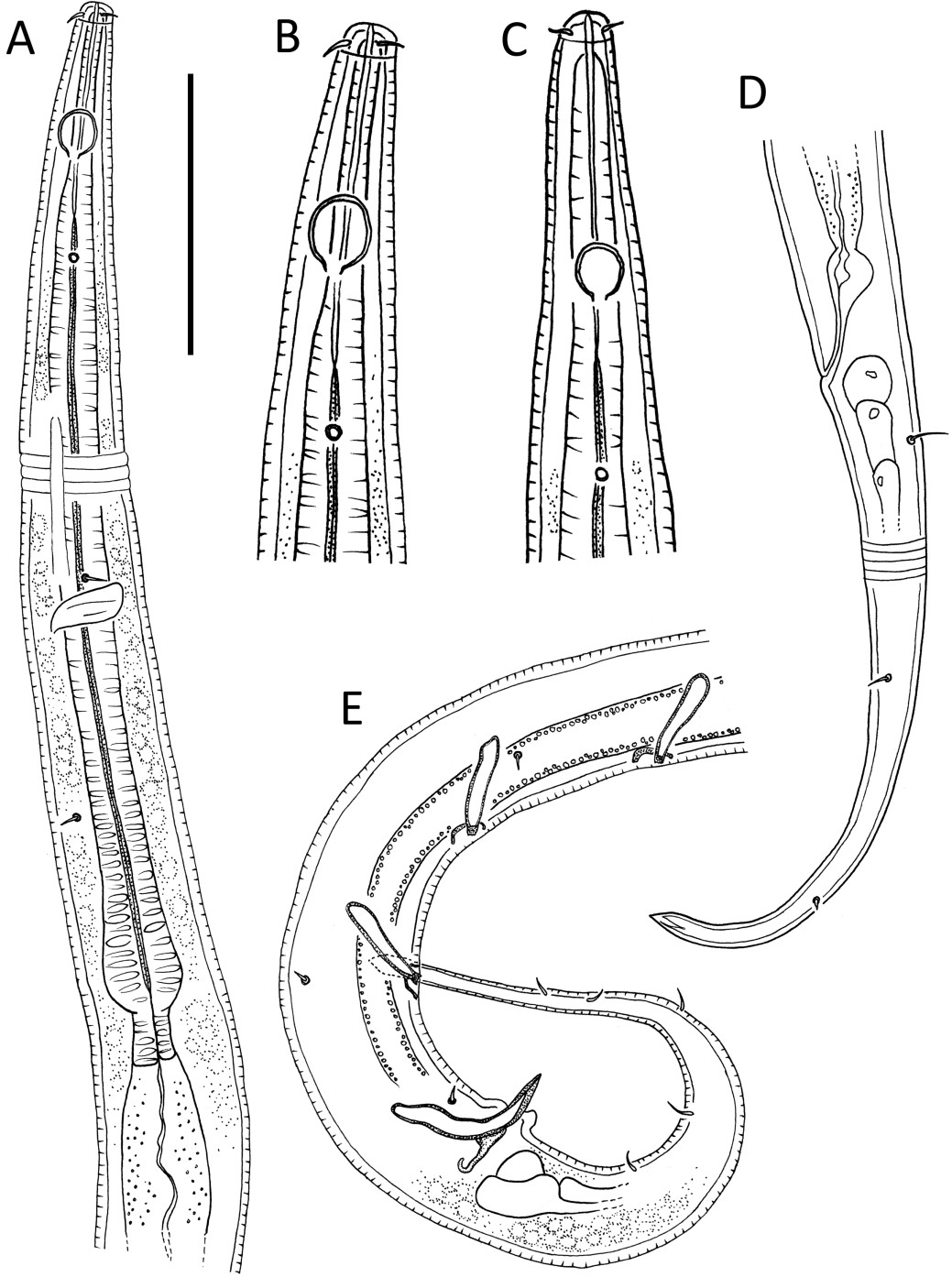

**Figure 1 *Leptolaimus hadalis* sp. nov.** (A) Anterior body region of male. (B) Male cephalic region. (C) Female cephalic region. (D) Female posterior body region. (E) Male posterior body region. Scale bar: A = 40 μm, B and C = 25 μm, D = 42 μm, E = 35 μm.

**Description:** Male. Body colourless, tapering slightly towards both extremities, with tail or posterior body region curved ventrally (maybe as a result of formalin fixation). Cuticle annulated, annules ca. 1.2 μm apart; lateral alae present consisting of raised,

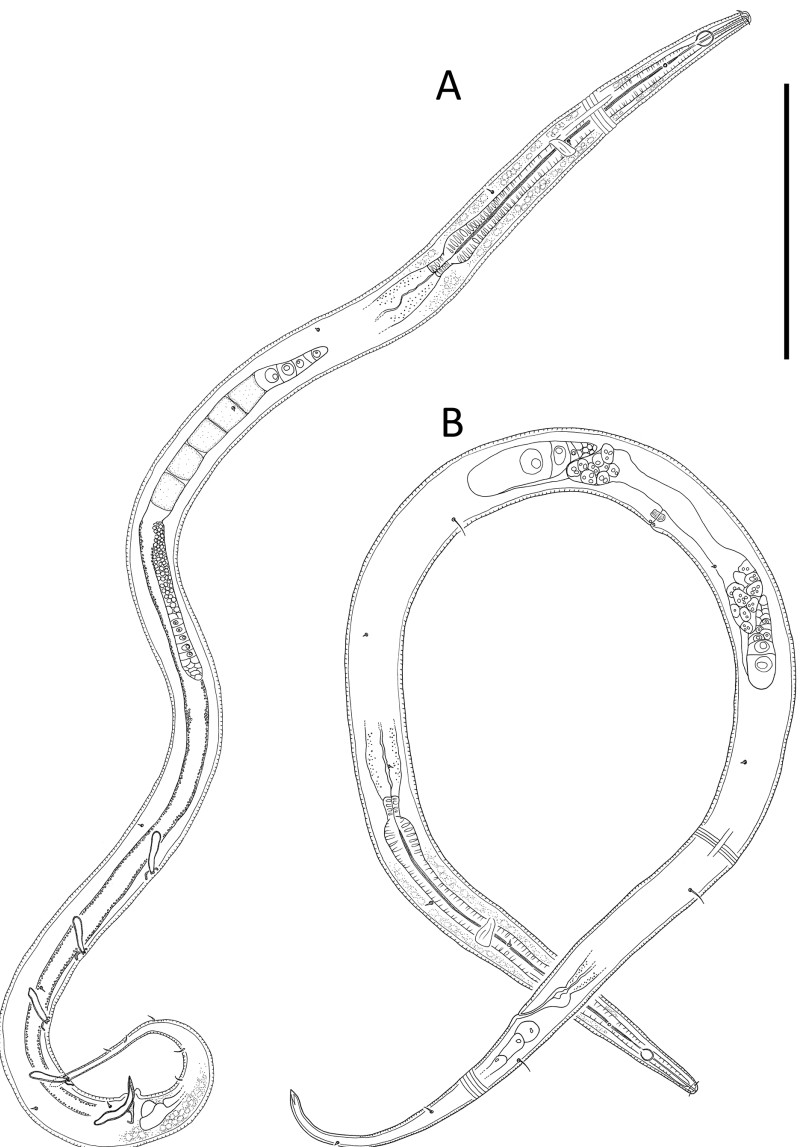

**Figure 2** *Leptolaimus hadalis* **sp. nov.** (A) Entire male. (B) Entire female. Scale bar = 100 µm.

non-annulated cuticle ca. 2–3 µm wide extending from posterior to first (anterior-most) body pore to anterior portion of tail. Four longitudinal rows of sublateral body pores, extending from near level of nerve ring to two thirds of tail length; anterior-most pore without seta, all other pores each with seta, 2–4 µm long; epidermal glands not observed. Sparse subventral and subdorsal somatic setae present on tail, 2–3 µm long. Labial region truncate to slightly rounded, not offset from body contour, lips fused. Inner and outer labial sensilla indistinct; cephalic sensilla setiform, 0.40–0.50 cbd long. Ocelli absent. Amphideal fovea round or slightly oval, located near middle or slighty posterior to middle of buccal cavity. Buccal cavity uniformly tubular: cheilostom and gymnostom short, undifferentiated; stegostom tubular, with uniformly thickened lumen. Pharynx muscular,

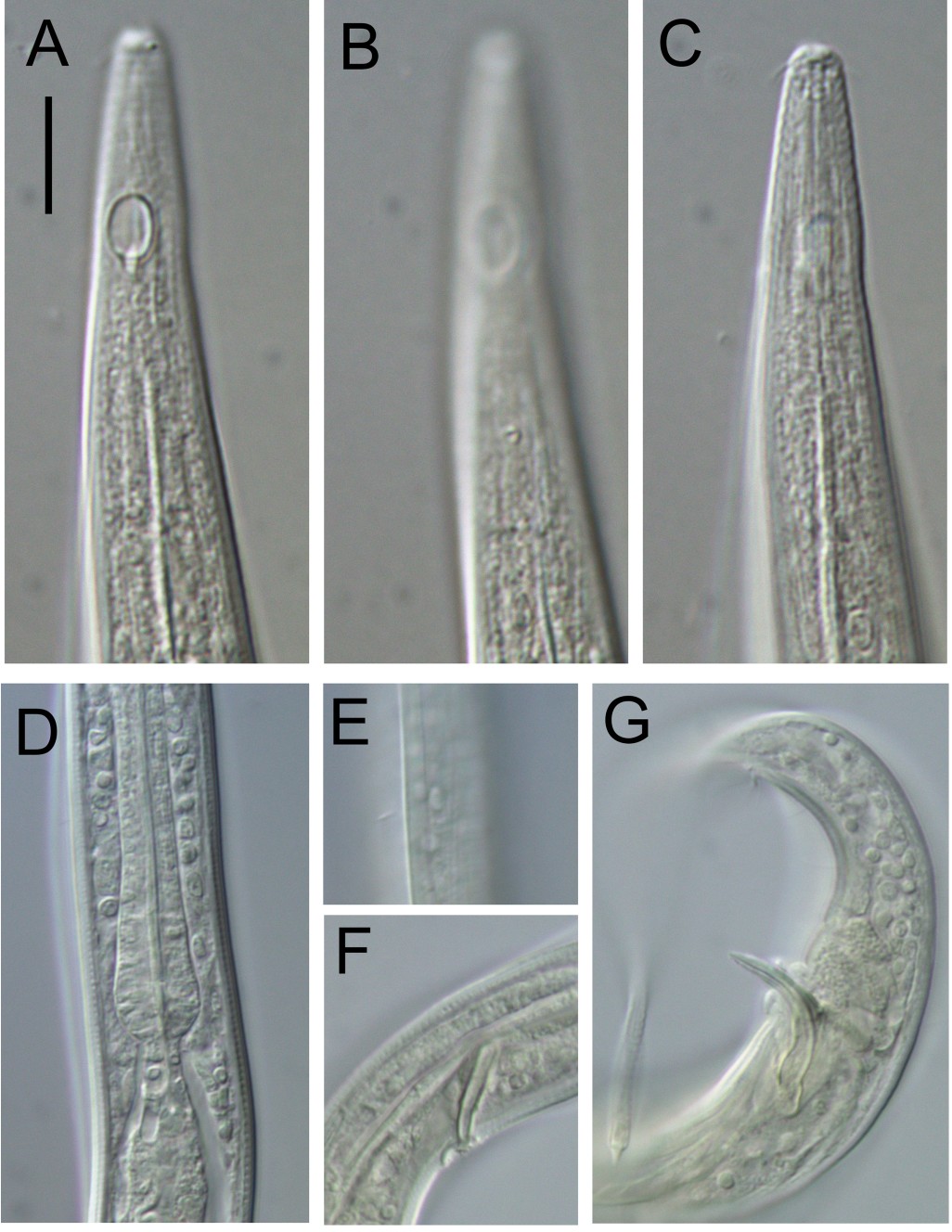

**Figure 3** *Leptolaimus hadalis* **sp. nov. Light micrographs of holotype.** (A–C) Anterior body region showing cuticle ornamentation, amphid, first body pore and buccal cavity. (D) Posterior pharyngeal region, cardia and intestine. (E) Mid-body cuticle with lateral ala. (F) Precloacal supplement. (G) Posterior body region with copulatory apparatus. Scale bar: A–C = 10 μm, D–G = 14 μm.

cylindrical anteriorly, with distinct oval basal bulb, slightly cuticularized lumen from posterior to buccal cavity to near posterior end of pharynx; valvular apparatus absent. Pharyngeal glands and their orifices indistinct. Nerve ring surrounding pharynx slightly

**Table 1 Morphometrics (μm) of *Leptolaimus hadalis* sp. nov. and *Alaimella aff. cincta* from Kermadec Trench.**

| | *Leptolaimus hadalis* sp. nov. | | | | | | | *Alaimella aff. cincta* |
| | Males | | | | Females | | | Male |
| | Holotype | Paratype 1 | Paratype 2 | Paratype 3 | Paratype 1 | Paratype 2 | Paratype 3 | |
|---|---|---|---|---|---|---|---|---|
| L | 638 | 612 | 650 | 741 | 681 | 587 | 628 | 1,508 |
| a | 34 | 32 | 34 | 37 | 28 | 27 | 30 | 52 |
| b | 5 | 5 | 5 | 5 | 5 | 5 | 5 | 7 |
| c | 6 | 7 | 6 | 7 | 6 | 6 | 6 | 11 |
| c′ | 6.0 | 5.0 | 6.3 | 5.6 | 7.9 | 7.8 | 8.7 | 6.2 |
| Head diam. at cephalic setae | 5 | 4 | 5 | 5 | 5 | 4 | 4 | 9 |
| Head diam. at amphids | 8 | 8 | 8 | 9 | 9 | 8 | 8 | 13 |
| Length of cephalic setae | 2 | 2 | 2 | 2 | 2 | 2 | 2 | 10–12 |
| Amphid height | 6 | 5 | 5 | 5 | 4 | 4 | 4 | 9 |
| Amphid width | 4 | 5 | 5 | 5 | 4 | 3 | 4 | 9 |
| Amphid width/cbd (%) | 50 | 63 | 63 | 56 | 44 | 38 | 50 | 69 |
| Amphid from anterior end | 14 | 14 | 15 | 16 | 19 | 12 | 15 | 4 |
| First body pore from anterior | 33 | 34 | 31 | 38 | 37 | 30 | 34 | – |
| Lateral alae from anterior | 49 | 49 | 41 | 71 | 64 | 54 | 58 | – |
| Buccal cavity length | 27 | 20 | 24 | 26 | 29 | 23 | 25 | – |
| Nerve ring from anterior end | 84 | 79 | 80 | 86 | 92 | 74 | 82 | 109 |
| Nerve ring cbd | 18 | 17 | 17 | 19 | 19 | 17 | 18 | 26 |
| Pharynx length | 134 | 124 | 127 | 138 | 140 | 112 | 132 | 221 |
| Pharyngeal bulb diam. | 10 | 10 | 10 | 10 | 10 | 10 | 10 | 12 |
| Pharynx cbd | 19 | 18 | 19 | 20 | 20 | 19 | 19 | 27 |
| Max. body diam. | 19 | 19 | 19 | 20 | 24 | 22 | 21 | 29 |
| Spicule length | 24 | 24 | 24 | 22 | – | – | – | 39 |
| Gubernaculum length | 9 | 11 | 8 | 7 | – | – | – | 11 |
| Cloacal/anal body diam. | 17 | 18 | 16 | 20 | 14 | 13 | 12 | 22 |
| Tail length | 102 | 90 | 101 | 111 | 110 | 101 | 104 | 137 |
| V | – | – | – | – | 348 | 282 | 308 | – |
| %V | – | – | – | – | 51 | 48 | 49 | – |
| Vulval body diam. | – | – | – | – | 24 | 22 | 20 | – |

posterior to middle of pharynx length. Secretory-excretory system not observed. Cardia cylindrical, 7–11 μm long, not embedded in intestine.

Reproductive system diorchic with opposed and outstretched testes; anterior testis located to the right of intestine with large, almost square sperm cells, $11 \times 11$ μm, posterior testis located to the right of intestine or ventrally, with small globular sperm cells, ca. $1.5 \times 1.5$ μm. Spicules paired, symmetrical, arcuate, 1.1–1.4 cloacal body diameter long, cuticularised; capitulum slightly rounded, shaft and blade gradually tapering distally. Gubernaculum with weakly cuticularized, dorsal apophysis with strongly curved distal portion; curved portion not always discernible. Ventral precloacal seta not observed.

Accessory apparatus composed of four, midventral tubular supplements ca. 19–38 μm apart, the two anterior-most supplements slightly further apart than the other supplements; tubular supplements 14 μm long, almost straight, slightly swollen proximally and with dentate tips, opening into shallow, cup-shaped, cuticularized depression in cuticle. Tail conico-cylindrical; three caudal glands and spinneret present.

Females. Similar to males, but with lower values of 'a', smaller amphids and longer tail. Reproductive system with two opposed reflexed ovaries; anterior ovary to the right of intestine and posterior ovary to the left of intestine. Spermatheca present in each genital branch. Vulva situated near mid-body. Vagina perpendicular, short, proximal portion encircled by single sphincter muscle and with slight cuticularisation distally. Vaginal glands not observed. Supplements absent.

**Diagnosis:** *Leptolaimus hadalis* sp. nov. is characterised by medium body 587–741 μm long, labial region not offset from body contour, inconspicuous labial sensilla, cephalic setae 0.4–0.5 μm long, buccal cavity 20–29 μm long, amphid located 12–19 μm from anterior end, female without supplements, male with four tubular precloacal supplements (alveolar supplements absent), tubular supplements almost straight with dentate tip, spicules arcuate, 22–24 μm or 1.1–1.4 cloacal body diameters long, and weakly cuticularized dorsal gubernacular apophyses strongly bent distally.

**Differential diagnosis:** The new species is most similar to *L. gerlachi Murphy, 1966*, *L. praeclarus Timm, 1961*, *L. nonus Holovachov & Boström, 2013* and *L. fluvialis Alekseev, 1981* in having relatively short (L < 1 mm) and plump bodies (a < 45), males without alveolar supplements and with continuous row of four tubular precloacal supplements, and females without tubular or alveolar supplements. *Leptolaimus hadalis* sp. nov. differs from *L. gerlachi* in shorter body length (587–741 *vs* 760–840 μm in *L. gerlachi*), slightly shorter cephalic setae (2 *vs* 3 μm), larger amphids (4–5 *vs* 3 μm wide in males), amphids located further posteriorly (12–19 *vs* 11 μm from anterior body extremity), first body pore located further posteriorly (30–38 *vs* 27 μm from anterior body extremity), longer buccal cavity (20–29 *vs* 18 μm), slightly shorter spicules (22–24 *vs* 28 μm) and tubular supplements with dentate tips (*vs* bifid tips). *Leptolaimus hadalis* sp. nov. differs from *L. praeclarus* in longer body length (587–741 *vs* 442–518 μm in *L. praeclarus*), longer tail (c′ = 5.0–8.7 *vs* 3.0–3.7), amphids located further posteriorly (12–19 *vs* 9 μm from anterior body extremity), first body pore located further posteriorly (30–38 *vs* 14 μm from anterior body extremity), shape of gubernacular apophyses (bent distally *vs* straight) and tubular supplements with dentate tips (*vs* bifid tips). *Leptolaimus hadalis* sp. nov. differs from *L. nonus* in having amphids located further posteriorly (12–19 *vs* 8–10 μm from anterior extremity), first body pore located further posteriorly (30–38 *vs* 17–29 μm from anterior body extremity), different lateral alae shape (raised cuticle *vs* two incisures), and shape of gubernacular apophyses (bent distally *vs* straight). *Leptolaimus hadalis* sp. nov. differs from *L. fluvialis* in having amphids located further posteriorly (12–19 *vs* 5–12 μm from anterior body extremity), shape of gubernacular apophyses (bent distally *vs* straight or slightly bent), and in the number of precloacal supplements (four *vs* six

supplements in original description). The two species also differ in their type habitat, *i.e.*, hadal trench *vs* lake sediments. The new species is also similar to *L. membranatus* (*Wieser, 1951*) *De Coninck, 1965* and *L. septempapillatus* Platt, 1973 in general body dimensions. *Leptolaimus hadalis* sp. nov. differs from *L. membranatus* in having four precloacal supplements (*vs* five in *L. membranatus*) and absence of supplements in females (*vs* one posterior tubular supplement in *L. membranatus*), and from *L. septempapillatus* by the shorter body length (587–741 *vs* 762–965 µm in *L. septempapillatus*) and the number of precloacal supplements (4 *vs* 7–8 in *L. septempapillatus*).

Family Camacolaimidae *Micoletzky, 1924*
Genus *Alaimella* *Cobb, 1920*

**Generic diagnosis: (from *Holovachov (2014)*)** Annules with fine longitudinal striations. Lateral alae absent. Somatic sensilla present. Ocelli absent. Amphideal aperture ventrally unispiral. Buccal cavity narrow, undifferentiated; cheilostom without cuticularisations; gymnostom undeveloped; stegostom linear, its lining continuous with that of *corpus*. Pharynx gradually widening posteriorly into a glandular "cylindrus". Female reproductive system monodelphic-opisthodelphic. Tubular and alveolar supplements absent. Spinneret weakly cuticularized.

**Type species:** *A. truncata* *Cobb, 1920*

**Remarks.** The genus comprises three valid species. A key to species was provided by *Tchesunov & Miljutin (2007)*.

***Alaimella aff. cincta*** *Cobb, 1920*
Figures 4–5, Table 1

**Material examined:** Adult male (NIWA 154901), edge of Kermadec Trench, 6,080 m water depth (*RV Tangaroa* voyage TAN1711, station 55, 32.1871°S, 176.5611°W), collected December 2017.

**Measurements:** See Table 1 for detailed measurements.

**Description:** Male. Body slender, tapering slightly towards both extremities. Cuticle with transverse annulations ca. 2.5 µm apart, beginning slightly anterior to amphids to near tail tip, with fine longitudinal striations. Short, sparse lateral somatic setae present in pharyngeal region, slightly anterior and posterior to level of nerve ring. Cephalic region not set-off from rest of body but narrowing slightly just anterior to amphids. Lip region poorly developed; inner and outer labial sensilla not observed. Four cephalic setae, 1.1–1.3 cbd long. Amphideal fovea relatively large, ventral unispiral (cryptocircular), with cuticularized outline; amphideal aperture of same overall dimensions as fovea but with circular outline. Buccal cavity narrow, undifferentiated; cheilostom without cuticularisation, gymnostom undeveloped, stegostom linear, its lining continuous with that of *corpus*. Pharynx muscular in anterior and middle portions, narrow in middle portion and widening into an elongated, glandular posterior bulb; nucleus of dorsal pharyngeal gland visible. Nerve

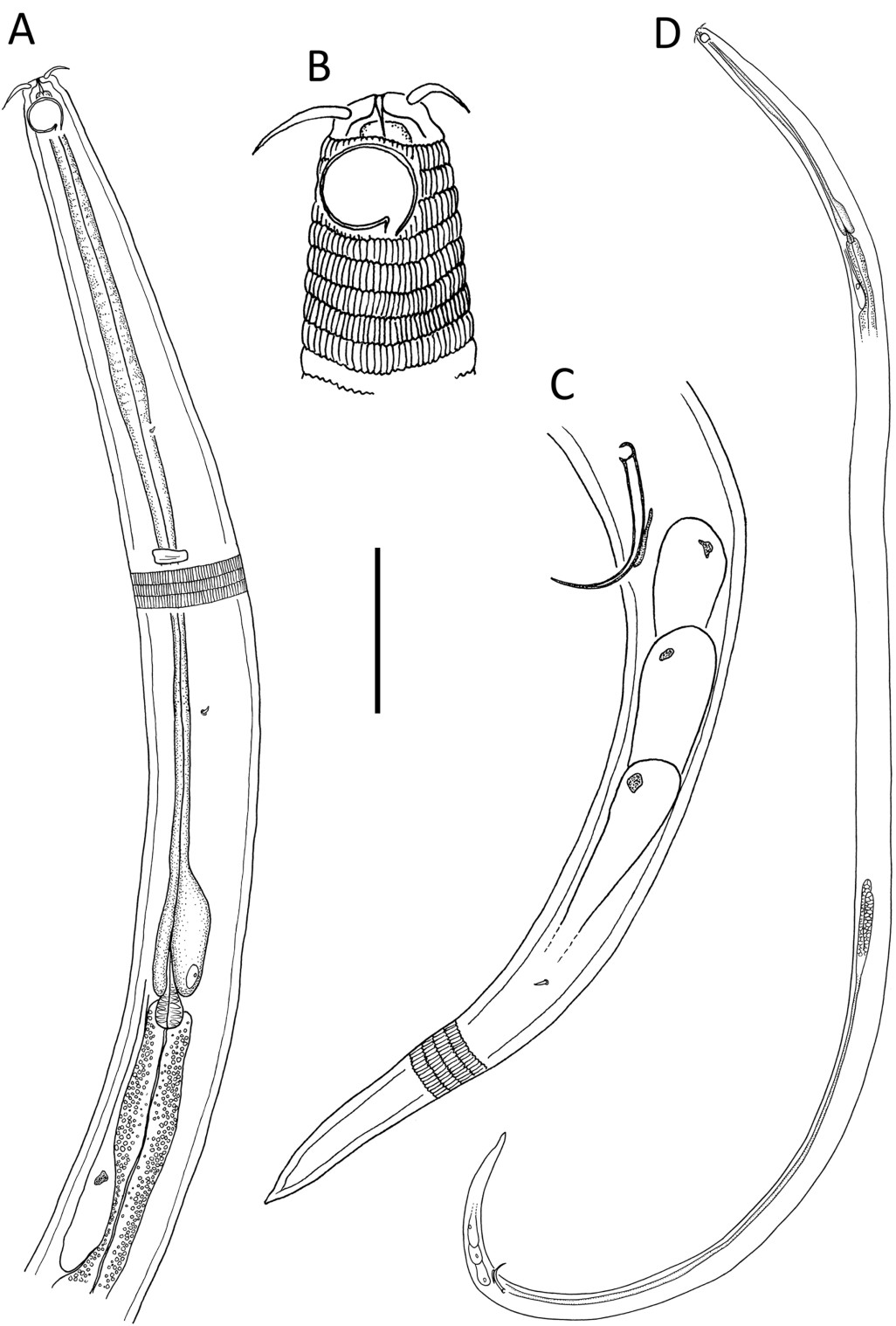

**Figure 4** *Alaimella aff. cincta* **male.** (A) Anterior body region. (B) Cephalic region. (C) Posterior body region. (D) Entire male. Scale bar: A = 40 μm, B = 12 μm, C = 28 μm, D = 125 μm.

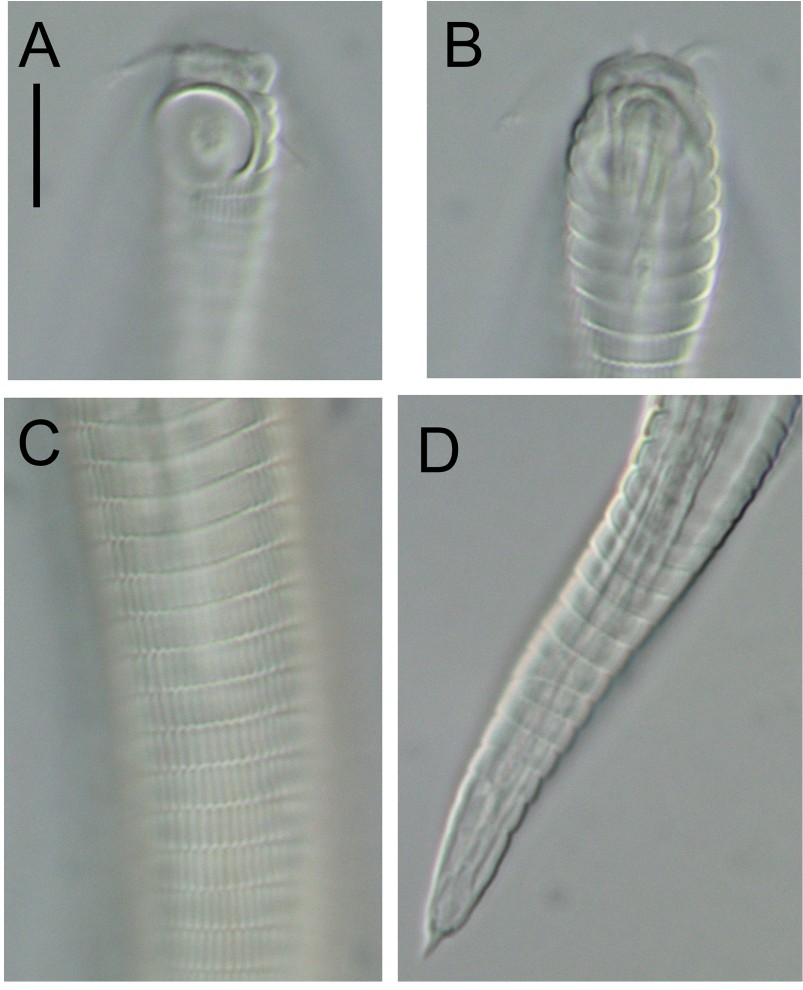

**Figure 5** *Alaimella aff. cincta* **Light micrographs of male.** (A and B) Cephalic region. (C) Cuticle of mid-body region. (D) Tail tip. Scale bar = 10 μm.     

ring located mid-pharynx. Cardia well-developed, ca. 7 μm wide and 10 μm long, surrounded by intestine. Secretory-excretory system present; ventral gland relatively large, located well posterior to pharynx, pore not observed.

Reproductive system diorchic with both testes relatively small, directed anteriorly and outstretched; both testes located ventrally relative to intestine. Sperm cells globular, ca. 2.5 × 2.5 μm. Spicules paired, symmetrical, slender, arcuate, 1.8 cloacal body diameters long; capitulum rounded, shaft and blade gradually tapering distally. Gubernaculum weakly developed, thin and plate-like. Precloacal supplements and ventral precloacal seta not observed. Tail conical, with pair of short lateral setae near two thirds of tail length from cloaca; three caudal glands present.

**Remarks.** The abyssal specimen agrees well with the original description of the species by *Cobb (1920)* based on specimens from Biscayne Bay, east coast of USA, including body length, cuticle ornamentation, size and shape of the amphids, length of cephalic setae, structure of pharynx, structure of male copulatory apparatus and tail. The abyssal

specimen is also very similar to the description of *L. cincta* by *Tchesunov & Miljutin (2007)* based on specimens from the subtidal zone of the White Sea, the only discrepancy being the larger amphids (9 µm wide or 0.7 cbd *vs* 5 µm wide or 0.6 cbd in the White Sea specimens).

Order Desmodorida *De Coninck, 1965*
Family Desmodoridae *Filipjev, 1922*
Genus *Desmodora de Man, 1889*

**Generic diagnosis: (modified from *Verschelde, Gourbault & Vincx, 1998*)** Cuticle without ridges or spines. Cephalic capsule either smooth or partly to entirely ornamented with structures resembling pores or small vacuoles, which have been shown by scanning electron microscopy to not be visible on the cuticle surface (*e.g.*, *Fadeeva, Mordukhovich & Zograf, 2016*); cephalic setae located either in the lip region or on main part of head capsule. Subcephalic setae sometimes present, when present few in number and mainly located posteriorly to amphideal fovea. Amphideal fovea cryptospiral or multispiral, seldom loop-shaped. Buccal cavity with large dorsal tooth and smaller subventral teeth. Pharynx with oval or circular posterior bulb. Spicules short, arcuate, with capitulum and velum. Precloacal supplements sometimes present, usually pore-like, seldom consisting of cuticular swellings or flaps. Tail usually conical, seldom conico-cylindrical.

Type species: *D. communis* (*Bütschli, 1874*)

**Remarks.** *Leduc & Zhao (2016)* provided a list of all 35 valid species of the genus. The exact nature of the rods or vacuole structures observed in the cephalic capsule of some species is yet to be determined with certainty using electron microscopy; these structures may in fact be rods in the median layer of the cuticle.

***Desmodora* aff. *pilosa* *Ditlevsen, 1926***
***= Desmodora gorbunovi* *Filipjev, 1946***
***= Desmodora gorbunovi perforata* *Filipjev, 1946***
Figures 6–8, Table 2

**Material examined:** Adult male, adult female and juvenile (NIWA 154902) from edge of Kermadec Trench, 6,080 m water depth (*RV Tangaroa* voyage TAN1711, station 55, 32.1871°S, 176.5611°W), collected December 2017.

**Measurements:** See Table 2 for detailed measurements.

**Description**: Male. Long cylindrical body, widest at level of pharynx, with slight golden-brown colouration throughout except for main portion of cephalic capsule which is strongly stained by Rose bengal; rounded anterior end and conical tail. Cuticle 3–4 µm thick, coarsely annulated with annulations slightly more widely spaced in pharyngeal region (ca. 1.5 µm apart) than on rest of body (ca. 1.1 µm apart). Eight longitudinal rows of somatic setae of variable length (4–10 µm) along entire body length. Well-developed

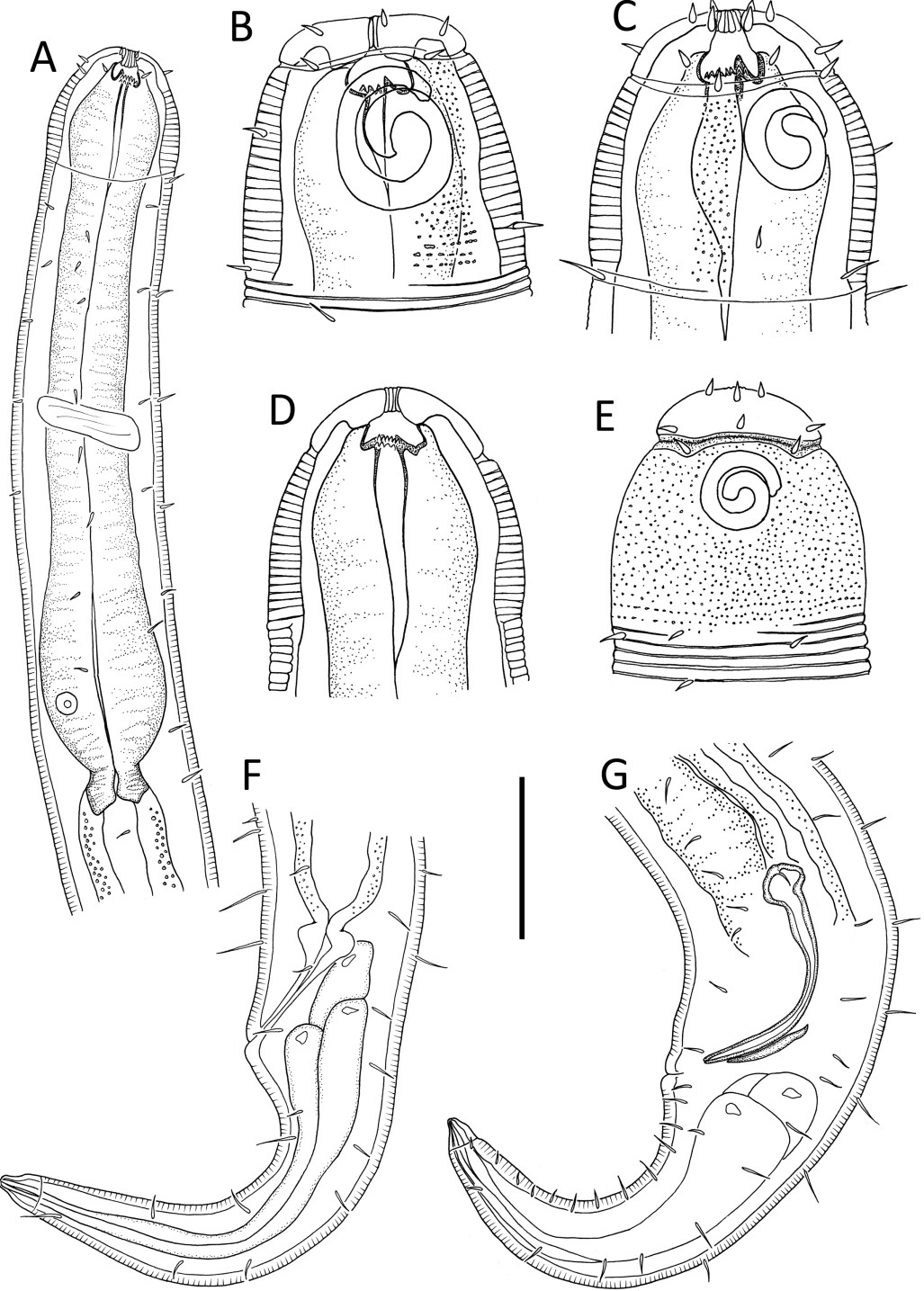

**Figure 6 *Desmodora* aff. *pilosa*.** (A) Anterior body region of female. (B) Male cephalic region. (C) Female cephalic region. (D and E) Juvenile cephalic region. (F) Female posterior body region. (G) male posterior body region. Scale bar: A = 60 μm, B–E = 25 μm, F and G = 45 μm.

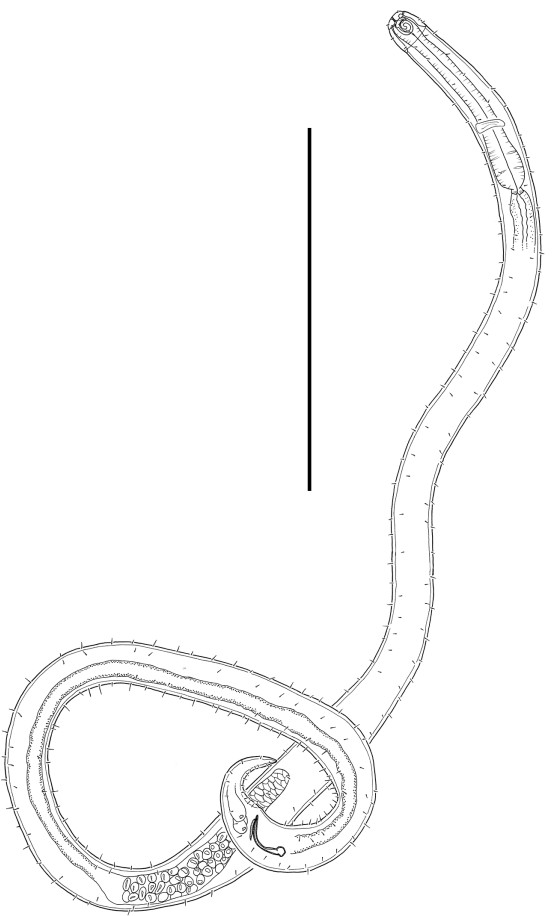

**Figure 7 *Desmodora* aff. *pilosa* Entire male.** Scale bar = 500 µm.

cephalic capsule, 44 µm wide and 45 µm long, consisting of two parts separated by a sutura: a lip portion with relatively thin cuticle and extendable anterior portion and a main region with thickened cuticle (up to 6 µm thick) and comprising at least three quarters of cephalic capsule. Main portion of cephalic capsule with numerous, dense pores (or vacuoles) irregularly distributed except near base where pores are arranged in transverse rows and sometimes merging with each other; short, 3–4 µm long, sparse subcephalic setae present at level of amphids or slightly posterior. Six inner and six outer labial setae present on lip region; inner labial setae 3 µm long, outer labial setae 4 µm long. Four short cephalic setae present at level of sutura, 0.2 cbd long. Amphideal fovea and aperture large, spiral with 1.5 turns and cuticularized outline, located on main portion of cephalic capsule; amphideal fovea slightly wider than amphideal aperture. Buccal cavity with large cuticularised dorsal tooth and two smaller ventrosublateral teeth; two lateral, transverse rows of small denticles also present. Cylindrical pharynx slightly swollen anteriorly and with oval posterior pharyngeal bulb. Secretory-excretory system not observed. Cardia 14 µm long, partially surrounded by intestine.

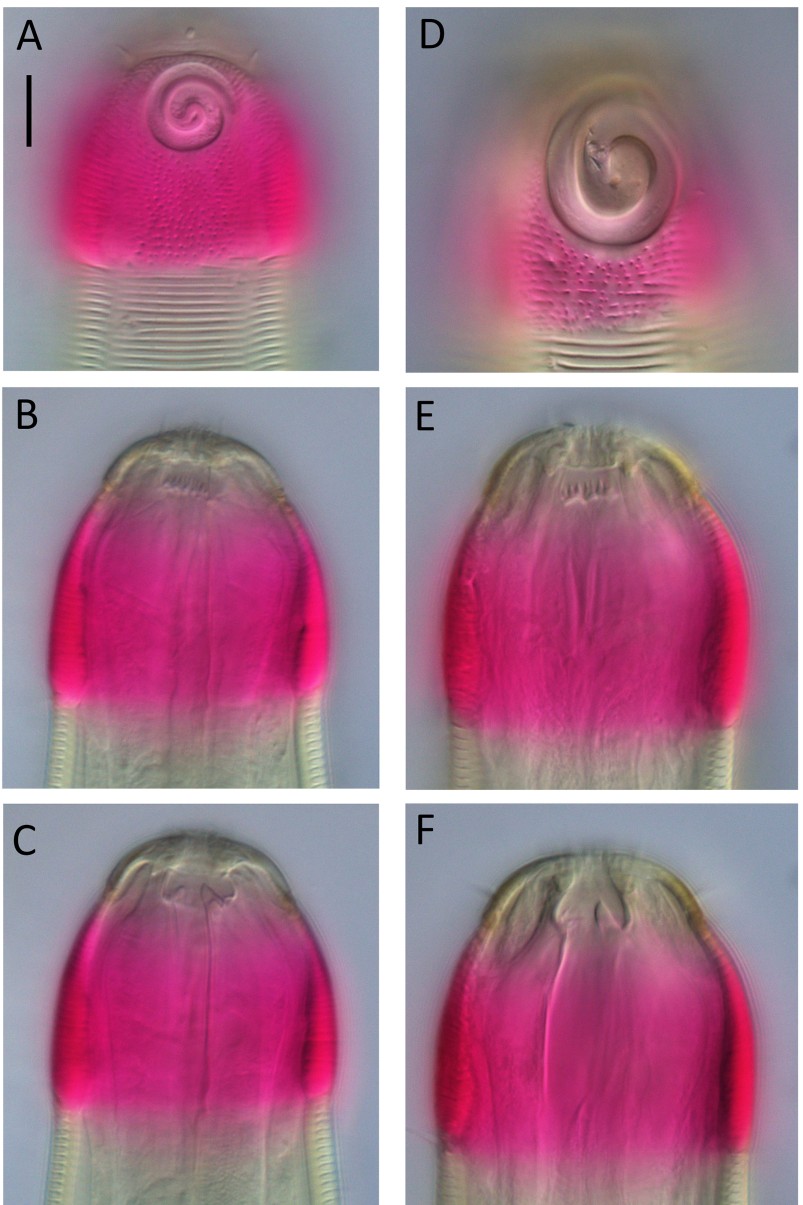

**Figure 8** ***Desmodora* aff. *pilosa* Light micrographs.** (A–C) Juvenile cephalic region. (D) Male cephalic region. (E and F) Female cephalic region. Scale bar = 10 μm.

Reproductive system monorchic with outstretched testis located to the right of intestine. Mature sperm cells globular to oval-shaped, 16–19 × 10–11 μm. Short, arcuate spicules with well-developed capitulum and tapering shaft and distal end; short, plate-like gubernaculum. Precloacal supplements and seta not observed. Conical tail with subventral and subdorsal rows of setae. Non-annulated tail tip without perforations. Caudal glands and spinneret present.

Female. Similar to males, but with smaller amphideal fovea with 1.25 turns and longer tail. Reproductive system didelphic, amphidelphic with reflexed ovaries both located to the

**Table 2 Morphometrics (μm) of *Desmodora aff. pilosa* from the edge of Kermadec Trench.**

| Species | *Desmodora aff. pilosa* | | |
|---|---|---|---|
| | Male | Female | Juvenile |
| L | 2,562 | 2,776 | 2,178 |
| a | 43 | 49 | 46 |
| b | 9 | 11 | 10 |
| c | 22 | 21 | 16 |
| c′ | 2.3 | 3.2 | 3.8 |
| Head diam. at cephalic setae | 28 | 32 | 28 |
| Head diam. at amphids | 39 | 42 | 34 |
| Length of sub-cephalic setae | 3–4 | 3–6 | – |
| Length of cephalic setae | 5 | 4 | 3 |
| Amphid height | 24 | 16 | 13 |
| Amphid width | 19 | 13 | 13 |
| Amphid width/cbd (%) | 49 | 31 | 38 |
| Amphid from anterior end | 7 | 11 | 10 |
| Nerve ring from anterior end | 176 | 136 | 131 |
| Nerve ring cbd | 55 | 58 | 52 |
| Pharynx length | 278 | 243 | 210 |
| Pharyngeal bulb length | 61 | 70 | 60 |
| Pharyngeal bulb diam. | 38 | 43 | 39 |
| Pharynx cbd | 54 | 56 | 53 |
| Max. body diam. | 60 | 57 | 47 |
| Spicule length | 75 | – | – |
| Gubernaculum length | 30 | – | – |
| Cloacal/anal body diam. | 51 | 41 | 36 |
| Tail length | 118 | 132 | 137 |
| V | – | 1,505 | – |
| %V | – | 54 | – |
| Vulval body diam. | – | 57 | – |

right of intestine. Spermatheca not observed. Vulva located slightly posterior to mid-body. Proximal portion of vagina heavily cuticularised; proximal portion of vagina surrounded by constrictor muscle. Vaginal glands present.

Juvenile. Similar to female, but with shorter body, smaller cephalic capsule, slightly shorter cephalic setae and slightly longer tail. Amphideal fovea with 1.5 turns.

**Remarks.** The Kermadec Trench edge specimens are broadly similar to previous descriptions of *D. pilosa*, although some discrepancies can be observed (Table 3). The structure and shape of the cephalic capsule, anterior sensilla, somatic setae, and amphideal fovea of the Kermadec Trench edge female are all similar to the original description of the Northwest Atlantic female by *Ditlevsen (1926)*. However the Kermadec Trench edge female specimen is characterized by shorter body (2,776 *vs* 3,000 μm), higher

**Table 3 Comparison of morphometric data of *Desmodora pilosa* Ditlevsen, 1926 from the literature.**

| | *Desmodora aff. pilosa* | | *Desmodora pilosa* | *Desmodora pilosa* | *Desmodora gorbunovi* and *D. gorbunovi perforata*[***] | |
|---|---|---|---|---|---|---|
| Reference | Present study | | Ditlevsen (1926) | Jensen (1991)[**] | Filipjev (1946) | |
| Location | SW Pacific | | NW Atlantic | Norwegian Sea | Arctic Ocean | |
| Water depth (m) | 6,080 | | 1,048–1,096 | 970–3,062 | 410–510 | |
| Sex | Male | Female | Female | Males | Males | Females |
| L | 2,562 | 2,776 | **3,000** | **2,820, 3,180** | **2,950–3,600** | **3,300–3,400** |
| a | 43 | 49 | **30** | **30, 34** | 45–50 | 35–40 |
| b | 9 | 11 | **13** | 8, 10 | 10–11 | 10–13 |
| c | 22 | 21 | **18** | **16, 19** | 18–22 | 19–21 |
| c′ | 2.3 | 3.2 | **3.5***| 2.5, 2.7* | 3.0* | 3.2* |
| Length of inner labial setae | 4 | 4–5 | ND | **6** | 5* | ND |
| Length of outer labial setae | 3 | 3 | ND | **9** | **6***| ND |
| Length of cephalic setae | 5 | 4 | ND | **9** | 5* | ND |
| Amphid height | 24 | 16 | ND | 21 | **17***| ND |
| Amphid width | 19 | 13 | ND | 20 | **17***| ND |
| Amphid width/cbd (%) | 49 | 31 | **37***| **23** | **37***| ND |
| Amphid turns | 1.5 | 1.25 | 1.25 | 1.25 | 1.25 | 1.25 |
| Buccal cavity armature | Dorsal + smaller ventrosublateral teeth and two lateral transverse rows of denticles | | ND | **Dorsal + smaller ventrosublateral teeth and band of denticles along ventral sector** | **Small dorsal tooth (ventrosublateral teeth may be present)** | |
| Spicule length | 75 | – | ND | **85, 96** | 72-80 | – |
| Gubernaculum length | 30 | – | ND | **36, 45** | **40** | – |
| %V | – | 54 | 56 | ND | – | **62–65** |
| Cephalic capsule with vacuoles? | Yes | Yes | **No** | Yes | **Variable**[#] | **Variable**[#] |

Notes:
Values and character states for males and females in bold do not overlap with the Kermadec Trench edge (SW Pacific) specimens.
ND, no data; *Estimated from figure; **Morphometric data of females not provided; ****Desmodora gorbunovi* Filipjev, 1946 and *D. gorbunovi perforata* Filipjev, 1946 were synonymised with *D. pilosa* by Gerlach (1963); [#]Pores present in *D. gorbunovi perforata* but absent in *D. gorbunovi*.

'a' ratio (49 *vs* 30), shorter tail (c′ = 3.2 *vs* 3.5), slightly smaller amphideal fovea (0.31 *vs* 0.37 cbd) and cephalic capsule with numerous pores (*vs* no pores). Unfortunately, Ditlevsen (1926) did not provide a description of the male, or of the buccal armature, which limits morphological comparisons.

*Desmodora gorbunovi* Filipjev, 1946 and *Desmodora gorbunovi perforata* Filipjev, 1946 were synonymized with *D. pilosa* by Gerlach (1963). The former is characterized by

cephalic capsule without pores whereas the latter is characterized by numerous pores in the cephalic capsule. Both of these are described by *Filipjev (1946)* as having a buccal cavity with a small dorsal tooth (although small ventrosublateral teeth may be present based on the drawings) without denticles. Clearly, *Gerlach (1963)* did not consider that the presence or absence of pores in the cephalic capsule as sufficient to justify separating the two populations into distinct species. In his redescription of *D. pilosa* based on both males and female specimens from the Norwegian Sea, *Jensen (1991)* noted strong similarities between his specimens and *Desmodora gorbunovi* including the cephalic capsule with numerous pores, size and arrangement of anterior sensilla and somatic setae, structure of the amphideal fovea, and tail shape. Jensen's specimens, however, have a buccal cavity with a band of numerous denticles, whereas Filipjev's specimens do not have denticles. The descriptions of *Filipjev (1946)* and *Jensen (1991)* also show some inconsistencies in body dimensions and size of anterior sensilla and amphideal fovea (Table 3).

The Kermadec Trench edge specimens differ from the descriptions of *Jensen (1991)* and *Filipjev (1946)* in buccal cavity armature, *i.e.*, lateral transverse rows of denticles *vs* band of irregularly-arranged denticles along ventral sector in Jensen's description and no denticles in Filipjev's description. The Kermadec Trench edge specimens are also smaller (body length 2,562–2,776 *vs* >2,800 μm) than both descriptions and differ in a number of body dimensions and size of cephalic sensilla and amphideal fovea (Table 3).

## DISCUSSION

The present study provides some of the deepest species records for the genera *Leptolaimus*, *Alaimella* and *Desmodora*. Prior to this study, the deepest record of a *Leptolaimus* species was from the DISCOL nodule field in the abyssal East Pacific (*L. formosus Bussau, 1993* at 4,174 m; *Miljutin et al., 2010*), although unidentified *Leptolaimus* species have been found at low densities (<1% of total nematode abundance) from as deep as 10,811 m in the Tonga Trench (*Leduc et al., 2016*), 8,189 m in the Puerto Rico Trench (*Tietjen, 1989*) and 6,300 m in the South Sandwich Trench (*Vanhove, Vermeeren & Vanreusel, 2004*). *Leptolaimus hadalis* sp. nov. was among the most common species observed in a survey of Kermadec Trench axis sites (*Leduc & Rowden, 2018*; referred to as "*Leptolaimus* D" therein). This species was the most abundant species in a core obtained at 8,079 m water depth (21% of total nematode abundance) and third most abundant species in a core obtained at 7,132 m (6% of total), while only one individual was found at 6,096 m depth, and none at 9,175 m depth (*Leduc & Rowden, 2018*). *Leptolaimus hadalis* sp. nov. was not found in cores obtained from the edge and deep axis of Tonga Trench (6,250 and 10,811 m depths; *Leduc et al., 2016*).

The rare marine genus *Alaimella* is seldom encountered in sediment samples. Prior to the present study, *Alaimella* had only been recorded from coastal locations (*Cobb, 1920*; *Raes et al., 2007*; *Tchesunov & Miljutin, 2007*) as well as from the Weddell sea (*Alaimella* sp. at unspecified depth between 200 and 2,000 m; *Vanhove, Arntz & Vincx, 1999*). The record of *Desmodora aff. pilosa* at 6,080 m depth is the deepest record of a *Desmodora* species to date. The previous deepest records were for *D. nybelini Allgén, 1954* from the abyssal Atlantic at 4,590 m depth and *D. striatocephala Tchesunov, 2008* from

the Southeast Atlantic at 5,450 m depth. Unidentified *Desmodora* specimens have been found as deep as 6,300 m in the South Sandwich Trench (*Vanhove, Vermeeren & Vanreusel, 2004*).

It remains to be ascertained whether the *Alaimella aff. cincta* and *Desmodora aff. pilosa* specimens from Kermadec Trench are in fact new species. Because the morphology of *Alaimella* is relatively simple and given the relatively limited variation in morphological characters among species of the genus, molecular sequence data will be required to determine the identity of the Kermadec Trench edge population. It seems likely that the abyssal specimen described here represents a cryptic species given the distance from the type locality (Biscayne Bay, USA) and much greater water depth. Similarly, given the differences observed between the present description of *Desmodora aff. pilosa* and those of *Ditlevsen (1926)*, *Filipjev (1946)* and *Jensen (1991)*, it seems likely that the Kermadec Trench edge specimens represent a different species. However, this cannot be ascertained until further data are obtained from the type locality (upper continental slope of NW Atlantic) to determine the structure of the buccal cavity armature, presence or absence of pores in the cephalic capsule, and morphology of males. However, despite the uncertainty regarding the identity of the Kermadec Trench specimens, the detailed morphological descriptions provided here will enable comparisons to be conducted in future studies of deep-sea nematode communities.

## ACKNOWLEDGEMENTS

I thank the co-voyage leaders Ronnie N. Glud and Ashley A. Rowden and science party of voyage TAN1711, and the officers and crew of RV Tangaroa, for their contribution to sample collection. I also thank Tim Shank, principal investigator of the HADES project, and to expedition leader Casey Machado, the officers, crew and scientific personnel of RV Thomas G.Thompson (voyage TN309), and ROV Nereus engineers and technicians. I am grateful to three anonymous reviewers for providing constructive criticisms on the manuscript.

## ABBREVIATIONS

**a**       body length/maximum body diameter
**b**       body length/pharynx length
**c**       body length/tail length
**c**       prime; tail length/anal or cloacal body diameter
**cbd**       corresponding body diameter
**L**       total body length; n, number of specimens
**V**       vulva distance from anterior end of body
**%V**       V/total body length × 100

### Funding

Funding was provided by NIWA's Coasts and Oceans Centre Research Programme 'Marine Biological Resources' and the programme 'Impact of resource use on vulnerable deep-sea communities' (CO1X0906). The voyages were funded by European Research Council Advanced Grant (ERC adG 2014 grant agreement number 669947) as part of the HADES-ERC trench project, with additional support from various national research programmes, and by the National Science Foundation (NSF-OCE1130712, 1130494 and 1131620) as part of the HADES project (HADal Ecosystem Studies). The funders had no role in study design, data collection and analysis, decision to publish, or preparation of the manuscript.

### Grant Disclosures

The following grant information was disclosed by the authors:
NIWA's Coasts and Oceans Centre Research Programme: CO1X0906.
European Research Council Advanced Grant (ERC adG 2014): 669947.
HADES-ERC Trench Project.
National Science Foundation: NSF-OCE1130712, 1130494 and 1131620.

### Competing Interests

The authors declare that they have no competing interests.

### Author Contributions

- Daniel Leduc conceived and designed the experiments, performed the experiments, analyzed the data, prepared figures and/or tables, authored or reviewed drafts of the paper, and approved the final draft.

### Field Study Permissions

The following information was supplied relating to field study approvals (*i.e.*, approving body and any reference numbers):

The collection of sediment samples was conducted under Special permit 666 to NIWA granted by New Zealand's Ministry for Primary Industries.

### Data Availability

The raw data is available in Tables 1 and 2 and the Figures.

All of the specimens have been lodged in the NIWA Invertebrate Collection (physical address: NIWA, 301 Evans Bay Parade, Hataitai, Wellington 6021, New Zealand).:

Leptolaimus hadalis sp. nov.:

Holotype male (NIWA 154899), paratype male 1 (NIWA 154900), paratype male 2 (NIWA 154900), paratype male 3 (NIWA 154900), paratype female 1 (NIWA 154900), paratype female 2 (NIWA 154900), paratype female 3 (NIWA 154900).

Alaimella aff. cincta *Cobb, 1920*:

Adult male (NIWA 154901)

Desmodora aff. pilosa *Ditlevsen, 1926*

Adult male (NIWA 154902), adult female (NIWA 154902), juvenile (NIWA 154902)

### New Species Registration

The following information was supplied regarding the registration of a newly described species:

Publication LSID: urn:lsid:zoobank.org:pub:47EF84F2-7B80-460E-BAF0-747FA9DDD447

Leptolaimus hadalis sp. nov. LSID: urn:lsid:zoobank.org:act:892D1646-DD68-4601-A12C-BA26A88E1B1A

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
