# Peer review of "New free-living nematode species and records (Chromadorea: Plectida and Desmodorida) from the edge and axis of Kermadec Trench, Southwest Pacific Ocean"

_PeerJ, doi:10.7717/peerj.12037_

## Round 0.1 · original submission · Minor Revisions

Reviewer 1 recommends adding a photo of the described species, since the drawing does not fully reflect reality, and the photo shows exactly how the animal looks in reality. This is useful for zoologists who will identify species in their samples.

Reviewer 3 has requested that you cite a specific reference. You may add it if you believe it is especially relevant. However, I do not expect you to include the citation, and if you do not include it, this will not influence my decision.

Reviewer 1 ·

Basic reporting

Manuscript is devoted to the description of the free-living deep sea nematofauna that is important topic in the frame of world biodiversity investigation programs. Manuscript is well supported with materials and literature. Figures are of a very good quality.

Experimental design

Research is within aim and scope of the journal. Although line drawings are perfect there is still the lack of at least light microscopical photographs. Adding light microscopy and scanning electron microscopy photos might greatly improve the manuscript.

Validity of the findings

no comment

Additional comments

The weakest point of the manuscript is the absence of photos of described species with the exeptions of Desmodora pilosa. I believe that this disadvantage could be easily corrected.
The second main remark concerns the structure of cephalic capsule of Desmodora (line 272). Author describe the presence of numerous dense irregularly distributed pores. Using scanning electron microscopy it was shown that cephalic capsule of Desmodorids is smooth. What is seen as pores are in fact some king of rods in the median zone of cuticle that strengthen cephalic capsule and seen as pores or dots under light microscope. The question is why are they found in some species and not found in others? There are may be several explanation that needs additional studies.
Editorial corrections
Line 75-77 Repetition of previous paragraph
Line 119-120 Leduc 2020 - no reference in the reference list
Line 226 Secretroy-excterory .... present. The noun is omitted.
Line 316 Filipjev 1946 and Jensen 1991 - no references in the reference list
Line 464 B. cephalic region; .... posterior body. C is omitted
Table 1. Line 1-4. all abbreviation is given in material and methods part.
Table 2. The same with previous.

Reviewer 2 ·

Basic reporting

no comments

Experimental design

no comments

Validity of the findings

no comments

Additional comments

Well written manuscript without any issues. Too bad there are so few specimens of Alaimella or Desmodora but it would be rather challenging to try and collect more material. I recommend to accept the manuscript, as I have no comments or corrections.

Reviewer 3 ·

Basic reporting

Clear and unambiguous, professional English used throughout.

The manuscript includes description of new nematode species Leptolaimus hadalis sp. nov.. from the edge (6080 m depth) and axis (7132 m) of Kermadec Trench, Southwest Pacific with some remarks about the genus. Alaimella aff. cincta and Desmodora aff. pilosa are recorded for the first time from the Southwest Pacific region. This manuscript includes description of new nematode species provides detailed morphological comparisons between other species/populations from other localities in order to assess the status of the Kermadec Trench populations.
The manuscript contains new, original data. All tables and figures are important and contain necessary information. The description of new species is correct and clear. Drawings are photos are very good quality. A new zoological taxon Leptolaimus hadalis sp. nov, is provided the name and LSID(s) and the publication LSID.
Literature references, sufficient field background/context provided.

Experimental design

Original primary research within Aims and Scope of the journal.
Research question well defined, relevant & meaningful. The nematode fauna of these deep environments remains very poorly known, and because of this we have little information on their ecology and of patterns benthic diversity within and across trench habitats.
The manuscript provides morphological descriptions of nematode species from Kermadec Trench, Southwest Pacific Ocean. The new information in the manuscript will also help future investigations of trench ecology.
Rigorous investigation performed to a high technical standard.
Methods described with sufficient detail & information to replicate.

Validity of the findings

The manuscript contains new, original data.

All underlying data have been provided; they are robust, statistically sound.

Conclusions are well stated, linked to original research question and limited to supporting results.

Additional comments

The nematode fauna of these deep environments remains very poorly known, and because of this we have little information on their ecology and of patterns benthic diversity within and across trench habitats.
The manuscript provides morphological descriptions of nematode species from Kermadec Trench, Southwest Pacific Ocean. The new information in the manuscript will also help future investigations of trench ecology.
The manuscript includes description of new nematode species Leptolaimus hadalis sp. nov.. from the edge (6080 m depth) and axis (7132 m) of Kermadec Trench, Southwest Pacific with some remarks about the genus. Alaimella aff. cincta and Desmodora aff. pilosa are recorded for the first time from the Southwest Pacific region. This manuscript includes description of new nematode species provides detailed morphological comparisons between other species/populations from other localities in order to assess the status of the Kermadec Trench populations.
The manuscript contains new, original data. All tables and figures are important and contain necessary information. The description of new species is correct and clear. Drawings are photos are very good quality. A new zoological taxon Leptolaimus hadalis sp. nov, is provided the name and LSID(s) and the publication LSID.
I suppose, differential diagnosis for a new species of Leptolaimus hadalis sp. nov. is not complete enough. Your differential diagnosis needs more detail. There is no comparison with the close species L. membranaceus (Wieser 1951) and L. septempapillatus Platt 1973.
Although your results are compelling, the data analysis should be improved. You should discuss the paper Fadeeva, N. P. & Mordukhovich, V. V. (2007) [New and known Leptolaimidae (Nematoda, Chromadoridae) species in the Sea of Okhotsk and the Sea of Japan.] Zoologicheskii Zhurnal, 86, 3-15. [in Russian], where the Pacific species of the genus Leptolaimus are described.

---

## Round 0.2 · accepted · Accept

In the revised version you took into consideration all comments and remarks. I recommend accepting your manuscript for publication in PeerJ.